# Tertiary Hyperparathyroidism in Diabetic Nephropathy: An Underrecognized Complication—A Narrative Review

**DOI:** 10.3390/biomedicines13112604

**Published:** 2025-10-24

**Authors:** Mirona Costea, Dana-Mihaela Tilici, Diana Loreta Paun, Vanda Roxana Nimigean, Sorin Constantin Paun, Rucsandra Elena Danciulescu-Miulescu

**Affiliations:** 1Doctoral School, University of Medicine and Pharmacy “Carol Davila”, 020021 Bucharest, Romania; mirona.costea@drd.umfcd.ro; 2Endocrinology Department, Bucharest University Emergency Hospital, 050098 Bucharest, Romania; diana.paun@umfcd.ro; 3Faculty of Medicine, University of Medicine and Pharmacy “Carol Davila”, 020021 Bucharest, Romania; vanda.nimigian@umfcd.ro (V.R.N.); sorin.paun@umfcd.ro (S.C.P.); rucsandra.danciulescu@umfcd.ro (R.E.D.-M.); 4Department of Oral Rehabilitation, Faculty of Dentistry, University of Medicine and Pharmacy “Carol Davila”, 020021 Bucharest, Romania; 5Bucharest Emergency Clinical Hospital, Calea Floreasca 8, 014461 Bucharest, Romania; 6Department of Diabetes, Nutrition and Metabolic Diseases, “Carol Davila” University of Medicine and Pharmacy, 020021 Bucharest, Romania

**Keywords:** parathyroid hormone (PTH), tertiary hyperparathyroidism (THPT), chronic kidney disease (CKD), diabetic nephropathy

## Abstract

Tertiary hyperparathyroidism (THPT) arises in patients with chronic kidney disease (CKD) as a consequence of prolonged secondary hyperparathyroidism and is marked by autonomous parathyroid hormone (PTH) secretion. In some cases, parathyroid hyperplasia persists even after successful renal transplantation, resulting in sustained PTH elevation and hypercalcaemia. These alterations contribute to bone loss, vascular calcification, and increased cardiovascular risk. Management includes medical therapy with calcimimetics or vitamin D analogues and surgical intervention via parathyroidectomy. However, optimal timing and treatment strategy remain uncertain. This review examines the pathophysiology, clinical manifestations, and current management paradigms of THPT, with an emphasis on areas that require further research and consensus.

## 1. Introduction

Tertiary hyperparathyroidism (THPT) is a condition mainly seen in patients with end-stage renal disease (ESRD), especially after prolonged secondary hyperparathyroidism (SHPT). It involves autonomous parathormone (PTH) secretion and ongoing hypercalcemia even after the initial cause is addressed—usually following successful kidney transplantation. THPT signifies a failure of the parathyroid glands to return to normal function, often due to parathyroid hyperplasia or nodular changes caused by chronic stimulation during SHPT.

Diabetic nephropathy, a microvascular complication of diabetes mellitus, has become the leading cause of ESRD globally. The increasing prevalence of diabetes and its renal complications has significantly shifted the epidemiological landscape of chronic kidney disease (CKD) and related mineral and bone disorders. Patients with diabetic nephropathy experience a combination of metabolic derangements that affect calcium, phosphate, vitamin D, and PTH homeostasis. These patients often develop SHPT earlier and may have a more rapid progression toward tertiary hyperparathyroidism due to more severe and persistent alterations in mineral metabolism.

Despite the well-recognized association between CKD and mineral bone disease, the specific incidence and clinical trajectory of THPT in individuals with diabetic nephropathy remain under-investigated. It is unclear whether diabetic nephropathy confers an independent risk for the development of THPT beyond its role as a cause of CKD. Some studies suggest that diabetic patients may have a blunted PTH response in early CKD stages, while others report a higher prevalence of parathyroid hyperplasia in diabetic ESRD populations. Furthermore, chronic inflammation, oxidative stress, and insulin resistance—hallmarks of diabetes—may further exacerbate parathyroid gland dysfunction.

The management of THPT becomes particularly challenging in diabetic individuals, who commonly present with comorbidities, vascular calcifications, and elevated cardiovascular risk, necessitating a personalized approach to medical versus surgical treatment. Parathyroidectomy, while often curative, may be associated with increased perioperative risk in patients with diabetes and cardiovascular disease. On the other hand, medical therapies such as calcimimetics may be less effective in cases of nodular or autonomous hyperplasia.

This narrative review aims to explore the incidence and clinical implications of THPT in patients with diabetic nephropathy, highlighting the unique pathophysiological mechanisms, risk factors, diagnostic challenges, and treatment considerations specific to this population. By synthesizing available evidence from clinical studies, guidelines, and expert opinions, we aim to delineate the scope of the problem and identify potential directions for future research and clinical practice improvement. A deeper understanding of how diabetic nephropathy influences the development and course of THPT may ultimately lead to better stratification of at-risk patients and optimization of therapeutic interventions.

## 2. Materials and Methods

Study design and search strategy: For this narrative review, a comprehensive literature search was conducted. We have used major medical databases, including PubMed and Google Scholar. The search primarily focused on articles published between 2019 and 2025. Keywords used in the search included “tertiary hyperparathyroidism”, “diabetic nephropathy”, and “chronic kidney disease”. Manual screening of the bibliographies of included studies and clinical guidelines was performed to uncover further relevant literature.

Eligibility criteria and study selection: We included clinical studies, guidelines, and systematic reviews. Priority was given to publications that addressed the pathophysiology, clinical presentation, diagnosis, and management. Exclusion criteria: animal/in vitro studies, non-relevant reports, pediatric population. Single case reports were also excluded.

Data extraction and synthesis: Relevant data, including study design, diagnostic definitions, clinical findings, and treatment modalities, were systematically extracted. The data were qualitatively synthesized and interpreted in the context of existing guideline recommendations. No meta-analysis was performed.

## 3. Diabetic Nephropathy and THPT–Pathophysiological Mechanisms

Diabetes mellitus, a chronic metabolic disorder marked by hyperglycemia, is associated with a wide range of systemic complications, among which renal involvement represents a major cause of morbidity and mortality. Current projections estimate that over 12% of the global population will be affected by diabetes by 2045, with type 2 diabetes accounting for the vast majority of the cases [1,2,3].

The natural evolution of diabetes mellitus is characterized by numerous microvascular complications, including diabetic nephropathy. This is the most common complication for type 2 diabetes mellitus and has become the main cause of ESRD globally [4]. Over 25–40% of patients with diabetes mellitus can experience renal damage after 20 years of disease [3].

Diabetic nephropathy is characterized by persistent albuminuria, measured at least twice within 3–6 months, and a decreased glomerular filtration rate (GFR). The ultimate result of nephropathy is ESRD [3,5]. Usually, the decreased GFR occurs in association with hypertension, accelerating the progression of renal deterioration [4].

It is important for clinicians to detect as early as possible the presence of diabetic nephropathy. Various biomarkers have been investigated, with albuminuria generally chosen as the primary tool for assessing renal function. The detection of albumin in the urine reflects the increased glomerular permeability to macromolecules. However, this is not a perfect way of screening because albuminuria has limitations—it is not sensitive to diabetic nephropathy, may not be present in the early stages, and it has larger variability [4]. The gold standard method is a kidney biopsy, an invasive procedure only available in specialized centers. The future screening method is expected to be more specific, like urinary microRNA markers [6].

The pathogenetic mechanisms of diabetic nephropathy are extremely complex and need further investigation. The well-described mechanisms of developing diabetic nephropathy are the accumulation of extracellular matrix, the vascular thickening and hyalinization, and the perturbations of intraglomerular hemodynamics. The intrinsic mechanisms responsible for these changes include advanced glycation products, activation of protein kinase C, and reactive oxygen species. High-glucose flux leads to advanced glycation end products (AGEs). Their presence plays a major role in the activation of the renin-angiotensin system (RAS). Increased renin secretion due to alterations in the juxtaglomerular apparatus ultimately leads to the production of angiotensin II. This molecule is responsible for adrenal aldosterone secretion, promoting sodium retention. As a result, arterial hypertension develops, exacerbating renal damage [5].

The switch between microalbuminuria, defined by 30–300 mg/day, and macroalbuminuria, defined by more than 300 mg/day, represents a critical stage for the evolution of diabetic nephropathy. This is a marker of severe renal dysfunction and represents a critical stage of the disease. It is accompanied by glomerular filtration rate deterioration and represents a sign of ESRD becoming imminent [7].

The long evolution of kidney disease frequently triggers additional metabolic disturbances, among which phosphate retention is particularly significant. Fibroblast Growth Factor 23 (FGF23) and its cofactor Klotho are important key regulators in phosphate metabolism. While the renal dysfunction continues, the level of FGF23 rises. FGF23 is the product of bone cells—osteocytes and osteoblasts and is responsible for decreasing plasma phosphate. This process is the result of reducing tubular phosphate reabsorption in a similar way to PTH. Also, FGF23 secretion decreased the renal synthesis of calcitriol, playing a crucial role in developing hypocalcemia. The underlying mechanism for FGF23 to activate its receptors is the presence of Klotho, a cofactor of the process. FGF23 alone cannot express its actions, as the activation of its receptors needs Klotho. Despite advances in understanding, the precise mechanisms governing the FGF23-Klotho axis remain incompletely elucidated and warrant further investigation [8,9].

In CKD patients, hyperphosphatemia associated with calcitriol deficiency leads to hypocalcemia, which in turn stimulates PTH secretion, ultimately resulting in secondary hyperparathyroidism. This form of hyperparathyroidism is characterized by adequate PTH secretation due to hypocalcemia. In the parathyroid glands, oxidative stress has been shown to disrupt calcium-sensing receptor signaling and the parathyroid cell proliferation control mechanism. Inflammatory cytokines can also modulate PTH gene expression and secretion, potentially accelerating the onset of secondary hyperparathyroidism. This highlights a more direct pathophysiological link between diabetes-related inflammation and parathyroid dysfunction, beyond the traditional CKD-mediated pathway [10].

Secondary hyperparathyroidism in the context of CKD often represents a transitional state. With disease progression, PTH secretion becomes persistent and dysregulated. This advanced stage is defined as THPT, characterized by autonomous parathyroid gland function, hypercalcemia, and progressive enlargement of all four parathyroid glands—Figure 1 [10].

The development of TPTH is multifactorial and primarily associated with CKD progression. The risk is considered to be maximal once the GFR drops below 45 mL/min/1.73 m^2^, reflecting significant loss of renal function and disrupted calcium-phosphate homeostasis. Several clinical and biochemical risk factors have been associated with the progression of TPHT. These are summarized in Table 1 below [11].

Given the overlapping clinical and biochemical features, distinguishing between secondary and tertiary hyperparathyroidism can be challenging. To aid clinicians in this differentiation, the following table, Table 2 summarizes the key distinguishing characteristics of these two entities [8].

PTH actions are modulated by the Calcium Sensing Receptor (CaSR), localized at parathyroid and kidney levels. It is capable of detecting slight variations in ionized calcium levels and stimulating PTH secretation when the ionized calcium level is low. PTH is a hypercalcemic hormone that acts on various organs—bone, kidney, and gastrointestinal tract. Via PTH Receptor 1, PTH increases tubular absorption of calcium, decreases phosphate concentration, and stimulates the activation of vitamin D. PTH also applies its actions on the bone and intestines, leading to calcium absorption [12].

THPT is a condition that affects the patient’s quality of life, in addition to the renal disease [13]. It is vital to have a multidisciplinary approach in managing THPT [7].

## 4. THPT and Kidney Transplant

The role of kidney transplantation in managing end-stage renal disease (ESRD) should be emphasized. The majority of patients with THPT are kidney transplant recipients who, despite restoration of renal function and interruption of the initial stimulus for PTH secretion, continue to exhibit elevated PTH concentrations. This persistence indicates autonomous parathyroid secretion, thereby characterizing TPTH [14].

After a successful kidney transplant, PTH and FGF23 levels should drop off in the first 3 months [15,16]. High levels of PTH after this period are responsible for hypercalcemia and hypophosphatemia and define THPT. Well-known risk factors for developing THPT after transplant are high PTH values before transplant, long-time dialyzed patients, and nodular parathyroid hyperplasia. Vitamin D in its active form also plays a role in the calcium and PTH axes. It is crucial to measure the levels of 25-OH-Vitamin D, as the deficiency of this metabolite may contribute to high PTH levels [16].

In a cohort study where 849 patients were followed 1 year after a kidney transplant, 21.5% of them developed THPT, defined by elevated PTH levels (>70 pg/mL) and hypercalcemia (>10 mg/dL). Pre-transplant PTH levels greater than 300 pg/mL were established to be a risk factor for developing TPHT. The study shows that CKD is a complex disease, and its effects can persist long after a renal transplant. Clinicians must be aware of this fact and ensure continuous patient monitoring [17].

Other data suggest even a greater percentage of patients suffering from THPT after renal transplant, up to 50% [18].

It is important to screen, recognize, and treat THPT, as it is a condition associated with renal allograft dysfunction and chronic allograft nephropathy [19]. THPT is also a risk factor for graft failure and ultimately death of the patience [20].

There is an urgent need to standardize the diagnostic approach. We must take into account biochemic findings—including high PTH, bone-specific alkaline phosphatase, vitamin D, and calcium levels– the preferred formula involves using the corrected calcium-to-albumin ratio [21].

To distinguish residual secondary HPTH from autonomous TPTH post-transplant, clinicians should consider the following practical criteria:Time course: persistent PTH elevation beyond 6–12 months post-transplant may suggest tertiary hyperparathyroidismCalcium levels: hypercalcemia supports the diagnosis of TPTH; in contrast, residual SPTH is typically associated with normocalcemia or even hypocalcemiaPhosphate levels: hypophosphatemia is more common in TPTH due to excessive PTH secretionPTH response to treatment: lack of improvement with correction of vitamin D deficiency and withdrawal of calcimimetics or phosphate binders may indicate autonomous secretion

Applying their criteria, clinicians can appropriately guide further diagnostic steps [11,14,16].

## 5. Clinical Manifestations

Chronic hypercalcemia presents with a wide spectrum of clinical manifestations. It can vary from no symptoms to severe bone affection. Non-specific symptoms include abdominal pain, headache, fatigue, irritability, and weakness. Pruritus may also be present in some cases. Bone disease may take the form of severe osteoarthritis, known as renal osteopathy [22]. Skeletal anomalies are found in the majority of stage 5 CKD patients [15].

The gold standard for diagnosis and assessment of renal osteodystrophy (ROD) remains bone biopsy, which allows histomorphometric analysis. The TMV classification (bone turnover, mineralization, and volume) categorizes ROD into osteitis fibrosa, adynamic bone disease, osteomalacia, and mixed uremic osteodystrophy [15]. Despite being the gold standard, bone biopsy is infrequently utilized in clinical practice due to its invasive nature and substantial cost. As a result, non-invasive imaging modalities such as 18 F-sodium fluoride positron emission tomography (18F-NaFPET) have been investigated and demonstrated strong correlations with histomorphometric indices. This investigation offers high diagnostic accuracy in the classification of ROD subtypes [23]. Unfortunately, this method is currently not available in Romania, limiting its clinical application in our country.

Osteitis fibrosa is the result of prolonged hyperparathyroidism [15,24]. PTH increases bone resorption and leads to negative bone balance. The osteoid formed under the elevated PTH status is not structurally normal, and it is called woven bone. The bones lose their physiological laminar disposition. Also, PTH is responsible for the appearance of peritrabecular fibrosis. In the end, cortical bone is thinned, and the laminar osteoid is replaced by fibrous tissue, accumulating cysts [24].

Adynamic bone disease is defined by low bone turnover and normal mineralization. It is the main bone affection among dialysis patients [15]. The activity of both osteoblasts and osteoclasts is severely reduced [25]. Adynamic bone disease manifests with bone pain and fracture. There are biochemical findings that are suggestive of adynamic bone disease: low levels of PTH, low bone-specific alkaline phosphatase, hypercalcemia, and vascular calcification, with the golden standard for diagnosis remaining bone biopsy. Bone turnover markers, such as bone-specific alkaline phosphatase, can aid in the assessment of bone metabolic activity [26].

Osteomalacia is often characterized by severe diffuse pain, in contrast to osteoporosis [25]. Osteomalacia is characterized by low bone turnover and abnormal mineralization. Its prevalence is now decreasing, as it was historically caused by the use of aluminum-based phosphate binders [15].

The metabolism of the skeletal system is different in CKD patients compared to the general population. Factors that contribute to the bone alteration in CKD patients are hemodialysis, peritoneal dialysis, or a history of kidney transplantation [24]. When discussing bone diseases, osteoporosis cannot be excluded. Osteoporosis is a condition characterized by low mineral density of the bones and is defined as a T-score below −2.5 SD. It is important to use DXA measurement for CKD patients, but we must consider the fact that it is not a helpful tool when wanting to capture the quality of the bone [24,27]. The bone mineral density is expected to be lower than in the general population, especially when measuring the density of the femoral neck and total hip. Factors involved in low mineral density are age, low body mass index, use of tobacco, and high PTH levels [24]. Although it has certain limitations, DXA remains a useful non-invasive tool for monitoring bone health in CKD patients. Bone quality can be measured in specific centers by using high-resolution quantitative peripheral computed tomography (HR-pQCT) [27].

It is well-known that fracture risk is significantly higher in patients with CKD than in the general population. Various studies have shown that the risk of fragility fractures is 5 times higher in patients with eGFR below 15 mL/min/1.73 m^2^ compared to individuals with eGFR higher than 60 mL/min/1.73 m^2^ [24]. Besides the fracture risk, bone disease is a well-known negative factor for developing cardiovascular comorbidities and mortality [15].

Chronic kidney-disease-related bone and mineral disease (CKD-MBD) is a complex clinical syndrome defined by a systemic disorder of mineral and bone metabolism due to CKD, which is manifested by abnormalities in bone and mineral metabolism and/or extraskeletal calcification. CKD-MBD arises as a result of the interaction among three distinct pathophysiological mechanisms: high PTH levels, abnormalities in bone metabolism, and diffuse soft tissue calcification. This may represent a potential background for patients to develop osteoporosis. Patients who develop this clinical entity are at high risk for fragility fractures, especially hip fractures [28]. The clinical findings include, besides bone affection, vascular calcification, neurological occurrences—dementia, cognitive decline, small vessel abnormalities, gastrointestinal events—constipation, liver inflammation, microbiome affection, infections, and malnutrition [29]. Several studies suggest that the development of CKD-MBD may be driven by a complex interaction between the immune system and the microbiome of the patients. This interplay serves as a foundation for ongoing research and the development of new therapeutic strategies [30].

While bone pain may be present in both diabetic neuropathy and TPTH, the latter is often associated with radiographic evidence of bone resorption, which is not characteristic of diabetic complications. Hypercalcemia in TPTH is typically persistent and biochemically distinct, whereas hypercalcemia is not a usual finding in diabetic neuropathy or nephropathy. Elevated PTH levels, in conjunction with hypercalcemia and imaging findings, support the diagnosis of TPTH [15].

Another important clinical aspect is related to the cardiovascular pathology of CKD patients. There are several risk factors for developing cardiovascular disease. They can be related to diabetes mellitus, such as insulin resistance, dyslipidemia, arterial stiffness, and arterial hypertension. Also, there are CKD-related mechanisms that can lead to cardiovascular injury due to the accumulation of uremic toxins, sodium retention, hyperphosphatemia, vascular calcifications, and elevated levels of renin and aldosterone [31]. Atherosclerosis is another important feature of this group of patients, and this can lead to severe events—stroke, coronary artery disease, or transient ischemic attack [21].

Soft tissue complications can also occur, with calciphylaxis (calcific uremic arteriolopathy) being the most severe manifestation. This affects small vessels, and it may trigger thrombosis, non-healing wounds, secondary infections, and skin necrosis, a potentially lethal condition [21].

## 6. Management Strategies

### 6.1. Medical Therapies

The optimal therapeutic approach varies according to the stage of the disease. If the patient is being carefully watched, the phase of hypocalcemia and hyperphosphatemia that will later trigger the secretion of PTH needs immediate and specific intervention [21]. Phosphorus levels typically increase when GFR goes under 30 mL/min/1.73 m^2^ [32].

Therapeutic options for CKD-MBD include vitamin D sterols, active vitamin D analogs, calcimimetics, and phosphate binders. Phosphate binders act by binding dietary phosphate in the gastrointestinal tract to prevent absorption and are classified as calcium-based or calcium-free. Calcium-based binders (calcium carbonate, calcium acetate) effectively lower serum phosphate but increase the risk of hypercalcemia, positive calcium balance, and vascular calcifications. Calcium-free binders (sevelamer hydrochloride/carbonate, lanthanum carbonate) are equally or slightly less effective but reduce the risk of hypercalcemia and vascular calcifications. However, they present specific adverse effects: sevelamer is associated with gastrointestinal intolerance, while lanthanum carbonate, though generally well tolerated, may lead to vomiting, tissue accumulation in bone and liver, and rare gastric mucosal deposition, with its long-term hepatic safety yet to be fully established [33].

The therapeutic response to agents such as calcimimetics, vitamin D analogues, and phosphate binders may offer important diagnostic clues in differentiating secondary from tertiary hyperparathyroidism. In secondary hyperparathyroidism, PTH levels usually respond to medical management—especially with calcimimetics, resulting in reduced PTH secretion. In contrast, THPT is characterized by autonomous PTH secretation, and patients often exhibit a minimal response to this therapy. Persistently elevated PTH and hypercalcemia despite optimal medical management may suggest the transition to THPT, thereby guiding consideration for surgical intervention. Recognizing these response patterns can aid clinicians in timely diagnosis and management [8,21].

The last line of therapy is dialysis, an efficient way to reduce phosphate, but difficult for the patient [21].

Sevelamer carbonate is a calcium-free agent designed to lower the levels of phosphorus. Besides this, it is also capable of binding bile acids and resulting in reduced LDL-cholesterol and preventing bone loss [32,34]. The target group for the use of this drug is patients who require dialysis or patients with phosphate levels higher than 1.78 mmol/L who do not require dialysis. The guidelines suggest a starting dose of 0.8–1.6 g/day, administered three times a day with meals. The typical adverse effects were related to the gastrointestinal tract—vomiting, nausea, dyspepsia, diarrhea, or constipation [34,35].

Although there are many therapies on the market, the perfect chelator has not yet been found—it should bind excess phosphorus in the gastrointestinal tract but without adverse effects and be cost-effective [32].

The involvement of cardiology is a critical component in the management of patients with CKD. Optimal pharmacological control of hypertension remains essential, with recommended target values below 120/80 mmHg. Anemia requires correction through iron supplementation or the administration of erythropoiesis-stimulating agents, aiming for hemoglobin concentrations of 10–12 g/dL in patients undergoing dialysis. Dyslipidemia also necessitates intervention, with therapeutic strategies including statins and sodium–glucose cotransporter-2 (SGLT2) inhibitors. Current guidelines additionally recommend the use of novel agents, such as monoclonal antibodies directed against dyslipidemia (e.g., alirocumab, evolocumab) and glucagon-like peptide-1 (GLP-1) receptor agonists [31]. Besides these facts, we must keep in mind that the risk of atrial fibrillation is higher in CKD patients and prevent embolic events by using anticoagulants [36].

### 6.2. Vitamin D Analogues

It was shown that 56% of the patients with stage 5 CKD lack vitamin D, a secosteroid hormone [37]. Vitamin D analogs target their receptors and contribute to the suppression of PTH synthesis [38]. Active forms of vitamin D include calcitriol, paricalcitrol, 22-oxacalcitriol, and others [37]. Synthetic calcitriol analogs include its prodrug, named Alfacalcidol, the most powerful and fastest analog for treating vitamin D deficiency at the moment. There are more analogs currently being the subject of various studies, like Falecalcitriol, which is potentially more efficient in PTH suppression [39].

These medications have the role of decreasing PTH secretion, but it was more recently shown that they also reduce the activation of the renin-angiotensin axis and reduce inflammation [37,40]. There is a proven reduction in PTH levels as a result of vitamin D analogs, but unfortunately, there are no well-defined target levels of PTH [41].

Several studies have failed to demonstrate a clear reduction in systemic calcifications and, in some cases, have even reported a potential increase in vascular calcification risk [38,42]. On the other hand, certain findings suggest that these interventions may exert favorable effects on vascular function; however, the evidence remains insufficient [40]. The role of vitamin D analogs in vascular calcification remains controversial. These conflicting data likely stem from differences in study design, dosing regimens, and patient characteristics such as the presence of diabetes and pre-existing calcifications. Moreover, the lack of standardized endpoints complicates comparison across studies. Further research is required to establish consistent, reliable biochemical targets and to clarify whether improvements in vascular function can occur independently of, or despite, progression in vascular calcification. Current guidelines offer limited and sometimes conflicting recommendations [38,40,42].

### 6.3. Calcimimetics

Calcimimetics represent a key category of drugs that revolutionized treating HPTH and CKD.

Calcimimetics function as allosteric modulators of the CaSR, a G protein–coupled receptor that governs PTH synthesis and secretion in accordance with fluctuations in extracellular calcium within the parathyroid glands. Practically, they imitate serum calcium’s effect on CaSR, inducing a decline in PTH secretation. They not only contribute to lowering PTH concentrations but also promote a reduction in parathyroid gland volume. [38]. Regardless of whether the patient is receiving concomitant vitamin D therapy or not, calcimimetics managed to reduce calcium and phosphate levels of patients with HPTH undergoing dialysis [43].

Currently, two orally administered calcimimetics—cinacalcet, the first agent of this class to receive approval, and evocalcet—are available, alongside an intravenously administered formulation, etelcalcetide [38,44]. Each of these three agents has its own disadvantages, and the choice of therapy should be individualized based on clinical and biochemical characteristics of the patient [38].

Cinacalcet was shown to have an anabolic effect on the bone, promoting bone formation and serving as a key ally in combating adynamic bone disease [45]. Cinacalcet is the most effective PTH-suppressing agent in the prevention of vascular events and bone fractures [36]. Its side effects are usually minor, including nausea and vomiting, but hypocalcemia may also occur. This remains the most prescribed calcimetic agent [44].

Evocalcet, the more recent oral calcimimetic therapy, has promising results and may become the first-line oral therapy. It has been shown to have better gastrointestinal tolerability compared to cinacalcet, but the incidence of hypocalcemia was similar [46].

Etelcalcetide, a novel second-generation intravenous calcimimetic, allows for thrice-weekly dosing during hemodialysis and was developed to improve treatment efficacy, patient adherence, and minimize gastrointestinal side effects compared with cinacalcet [47].

Etelcalcetide is more effective than cicalcet in lowering PTH levels. This fact tends to induce a higher rate of hypocalcemia (calcium <7.5 mg/dL). Beneficial effects on bone structure and a potential reduction in fracture incidence were described when using etelcalcetide [48]. It is crucial to mention that the progression of left ventricular hypertrophy was inhibited by the use of etelcalcetide, demonstrating cardiovascular benefits; however, further studies are needed in this area [49].

Upacicalcet is one of the new injectable calcimimetic agents, currently being used only in Japan. It is used for patients undergoing dialysis, being injected three times a week into the venous side of the dialysis machine. However, it is important to note that most of the evidence supporting the use of Upicalcet comes from early-phase clinical trials and observational data, predominantly conducted in Japan. Furthermore, long-term safety and efficacy data are still limited, especially in diverse patient populations. Thus, while the results available at the moment appear promising in terms of safety and tolerability, particularly the lower incidence of hypocalcemia, the overall quality of evidence remains moderate, and further validation is required [50]. It is considered to be effective and safe, with a low rate of induced hypocalcemia, in only 2% of patients from a 2023 study [51].

### 6.4. Surgical Interventions

Despite the availability of numerous pharmacological therapies, some patients persist with significantly high levels of calcium and PTH. Furthermore, even one year following renal transplantation, some patients do not attain adequate biochemical control, highlighting the potential necessity for radical surgical intervention [52].

At present, no standardized criteria have been established to define the optimal timing of surgical intervention. In clinical practice, PTH levels ranging from 2- to 9-fold above the upper reference limit are generally regarded as indicative of surgical necessity [10,16].

Although no universally standardized criteria exist, several clinical and biochemical thresholds are widely accepted in guiding surgical referral. Current guidelines, such as KDIGO and the European Society of Endocrinology, suggest considering surgery in the following scenarios:PTH levels persistently elevated at 2–9 times the upper normal limit despite optimal medical therapyChronic hypercalcemia, particularly if >11 mg/dLPersistent hypercalcemia for 3 to 12 months post-renal transplantHyperphosphatemia refractory to phosphate bindersSevere bone diseaseUncontrolled symptoms, such as pruritus, fatigue, nephrolithiasis, or nephrocalcinosis

The main goal of surgery is to reestablish normal calcium concentration while reducing parathyroid gland mass [16,53,54].

According to the Miami criteria, a successful surgical intervention is indicated by an intraoperative PTH decrease of over 50% measured at least 10 min post-resection [16,55]. Other authors consider that a reduction in PTH levels >7-% 20 min after surgery is a good indicator for a successful surgical act [56]. It is important to mention the difference in clearance of intact PTH in CKD patients, as it is much slower. In renal failure, the iPTH decay is around 6.6 min. If the levels of PTH do not drop as expected, the surgeon must take into consideration the existence of ectopic parathyroid glands [55].

It is recommended to use imagistic methods in order to localize the parathyroid glands. The routine use of parathyroid imaging remains a topic of debate. We suggest that imaging should not be routinely performed in all cases but rather reserved for those being considered for surgery or in whom the biochemical diagnosis is uncertain. The most accessible approach is by using ultrasonography—an evaluation best used in cases of over 1 g adenomas. Besides that, we can detect the presence of all four hyperplastic parathyroid glands by using Tc−99 m sestamibi SPECT-CT. Clinicians must guide the use of imagistic methods and adjust them to every patient’s needs [52]. On the other hand, some authors suggest that it is not mandatory to localize the parathyroid glands before surgery, and the Miami criteria are the most efficient indicator, as an experienced surgeon can find ectopic glands without preoperative localization [57]. Preoperative localization remains a useful tool, especially in cases of recurrent HPTH [56].

Globally, several operative approaches are implemented in the treatment of THPT. These include: subtotal parathyroidectomy with bilateral cervical thymectomy, entailing removal of three glands and partial excision of the fourth, with a residual parathyroid remnant of approximately 40–80 mg; total parathyroidectomy, with or without autologous parathyroid tissue transplantation, usually in conjunction with bilateral cervical thymectomy; and total parathyroidectomy without autotransplantation or associated thymectomy [16]. Thymectomy may be used, as the majority of ectopic parathyroid glands are situated there [56]. Potential sites for reimplantation include the sternocleidomastoid muscle, the brachioradial muscle, or, more appropriately, the subcutaneous adipose tissue of the forearm [16,56].

Total parathyroidectomy is generally not considered a first-line approach due to the high likelihood of chronic hypoparathyroidism, although it offers the benefit of a low recurrence rate. In contrast, subtotal parathyroidectomy is associated with a higher risk of disease recurrence, but postoperative hypocalcemia occurs less frequently. Total parathyroidectomy combined with autotransplantation has been shown to more effectively improve patients’ quality of life, although there is the potential for regrowth of the transplanted parathyroid tissue. According to some authors, this procedure remains the treatment of choice [56].

Given the complexity of the cervical region, various intraoperative assistive methods were developed to facilitate the identification of parathyroid glands. We mention noncarbon suspension-assistance with negative parathyroid imaging, near-infrared fluorescence, and the use of indocyanine green [56]. The technique of injecting indocyanine green dye followed by detection using fluorescence has been used for years in several centers. This proved to be safe, effective, and had a key role in the preservation of the parathyroid glands [58].

As surgery practice moves toward minimal methods, we have to mention ablation. This is a new way to approach tumors, and whether it is using microwave or radiofrequency, some studies show benefits when used for HPTH. While ablation techniques (including microwave and radiofrequency ablation) offer a minimally invasive alternative for patients with contraindications to surgery, most available data are derived from retrospective studies or small prospective cohorts [38,56]. This technique offers a valid alternative for patients with contraindications for anesthesia. Although a meta-analysis found no significant differences between parathyroidectomy and ablation in terms of PTH, calcium, or phosphate levels. Ablation has shown a lower rate of hypocalcemia after the procedure, a shorter hospitalization time, but an increased risk of recurrence. The heterogeneity of patient selection, procedural protocols, and outcome measures limits the generalizability of the findings [38]. In addition, the risk of recurrence and the lack of long-term follow-up in many studies suggest that ablation should currently be considered as an alternative option in selected cases, rather than a standard first-line therapy. The overall quality of evidence is moderate, emphasizing the need for randomized, controlled, and adequately powered studies [58].

The most clinically relevant complications of parathyroidectomy are hypocalcemia and hungry bone syndrome, a condition that can become lethal. Among patients undergoing dialysis, hungry bone syndrome has a high prevalence, ranging from 26% to 95%. This condition usually presents with a rapid onset of hypocalcemia and often lasts more than four days after surgery. The standard treatment includes continuous calcium infusion [59]. Elevated levels of PTH pre-operatively, with a cutoff of over 166 pmol/L, are a powerful predictor of post-operative hypocalcemia and require higher doses of replacement [60].

Besides this, hypoparathyroidism may appear as transitory or permanent, and it is accompanied by hypocalcemia [52]. Hypocalcemia may be asymptomatic or present as tingling and numbness of the extremities. Clinical indicators of hypocalcemia include tetany, a positive Trousseau or Chvostek sign, and a prolonged QT interval. In cases of permanent hypoparathyroidism, long-term complications may arise, such as cataracts, cardiovascular involvement, and basal ganglia calcification. The treatment involves chronic calcium substitution, although ongoing research is exploring more effective therapeutic alternatives [61].

## 7. Conclusion and Future Directions

THPT complicating diabetic nephropathy is marked by a pronounced clinical heterogeneity and poses a multifaceted therapeutic challenge. The absence of standardized diagnostic criteria delays identification and consistent treatment selection. Management must therefore integrate targeted metabolic and cardiovascular measures—rigorous blood-pressure control, correction of anemia, treatment of dyslipidemia—alongside therapies directed at mineral metabolism, including vitamin D analogues, calcimimetics (cinacalcet, etelcalcetide, upacicalcet) and phosphate-lowering strategies; these interventions can favorably modify PTH levels, parathyroid gland volume and bone structure, but require careful monitoring for hypocalcemia and vascular calcification.

Patients who remain refractory to medical therapy may benefit from surgical approaches (subtotal or total parathyroidectomy with or without autotransplantation) or minimally invasive ablation. However, these options also carry risks—such as hungry bone syndrome or persistent hypocalcemia, demanding meticulous perioperative planning.

Future research should prioritize the identification and validation of specific biomarkers that can better stratify disease severity, predict therapeutic response, and monitor disease progression in SPTH and TPTH. At the moment, there is a lack of validated, non-invasive biomarkers beyond serum PTH and calcium-phosphate levels, which show high variability and limited predictive value.

Moreover, consensus is still lacking regarding optimal PTH targets in dialysis patients, with current guidelines offering wide reference ranges without solid evidence for improved outcomes. There is a critical need for large-scale, longitudinal studies to determine the PTH thresholds that correlate with reduced cardiovascular calcification and improve bone turnover, particularly in high-risk subgroups such as diabetic patients.

In addition, further investigation into the long-term cardiovascular and skeletal outcomes of emerging pharmacologic agents is needed, along with personalized approaches that account for comorbidities such as diabetes.

Ultimately, developing individualized treatment algorithms that integrate pharmacologic, interventional, and surgical modalities—tailored to patient phenotype and comorbidity profile- will be essential to improving quality of life and long-term survival in this complex patient population.

## Figures and Tables

**Figure 1 biomedicines-13-02604-f001:**
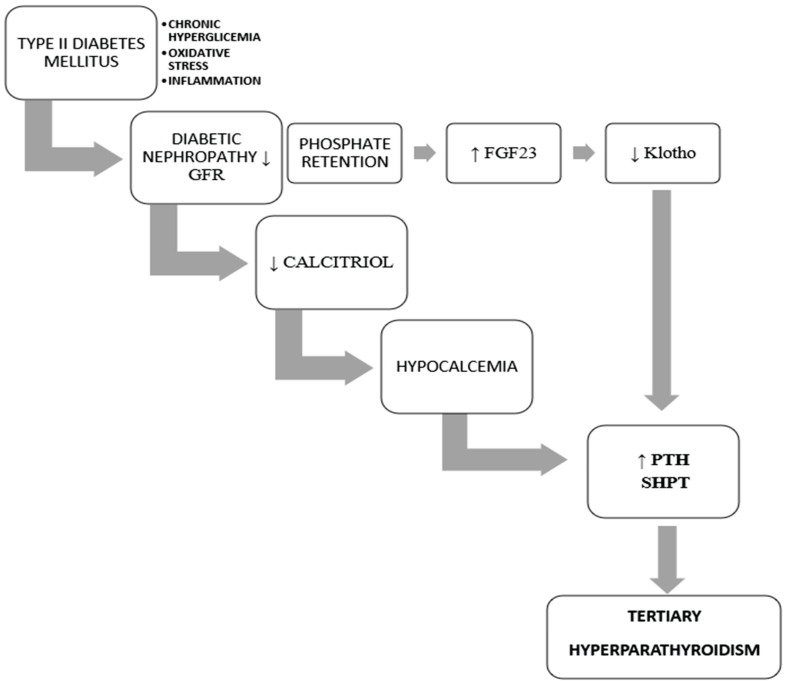
Pathophysiological Mechanism Leading to THP.

**Table 1 biomedicines-13-02604-t001:** Summary of risk factors associated with TPTH.

Risk Factor	Mechanism
eGFR < 45 mL/min/ 1.73 m^2^	Reduced phosphate excretion, vitamin D deficiency
Young age	Higher bone turnover and sensitivity to mineral imbalance
Male sex	Possibly linked to differences in skeletal and hormonal profiles
Hypoalbuminemia	Affects calcium binding and PTH regulation
Use of diuretics	May increase calcium excretion and contribute to chronic PTH stimulation
Long dialysis duration	Prolonged SPTH
Poor phosphate control	Hyperphosphatemia promotes PTH secretion
Vitamin D deficiency	Leads to hypocalcemia and compensatory PTH increase

eGFR = estimated glomerular filtration rate; PTH = parathormone; SPTH = secondary hyperparatyroidism.

**Table 2 biomedicines-13-02604-t002:** Distinguishing SHPT and THPT.

	SHPT	THPT
Biochemical Assessment	↑ PTH N/↓ Calcium	↑ PTH ↑ Calcium
Clinical Context	Low CKD stage	Renal transplantation history, chronic dialysis
Evaluate response to calcimimetics and vitamin D analogs	Responsive	Poor response
Imaging	No visible parathyroid adenoma	Parathyroid adenoma
Management	Monitor biochemical markers Optimize medical therapy	Optimize medical therapy Refer to surgery

N = Normal Level; SHPT = Secondary hyperparathyroidism; THPT = Tertiary hyperparathyroidism; PTH = parathormone.

## Data Availability

No new data were created.

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
