# Peer review of "Tertiary Hyperparathyroidism in Diabetic Nephropathy: An Underrecognized Complication—A Narrative Review"

_biomedicines, 2025, doi:10.3390/biomedicines13112604_

Round 1

Reviewer 1 Report

Comments and Suggestions for Authors

All comments and revisions have been provided within the manuscript. 

This narrative review tries to address different aspects of tertiary hyperparathyroidism in diabetic nephropathy. 

Despite having an interesting and potentially practical topic, the review article has major concerns and problems. For example, it lacks clear flow and hierarchy.  Generally, the manuscript has been written quite poor. Several paragraphs with no significant connections and no direct relevancy to the topic have been provided within the manuscript. In other words, the review is hectic and not organized well. On the other hand, possible differences of tertiary hyperparathyroidism in the setting of diabetes nephropathy with other clinical contexts of CKD (like hypertension, glomerulonephritis) regarding mechanism, signs/symptoms, diagnosis, and treatment approaches have not considered at all.  

Comments on the Quality of English Language

The manuscript has several grammatical and spelling errors. 

Some sentences do not make sense and are mostly vague.  

Author Response

Dear Reviewer.

We would like to begin by expressing our most sincere gratitude for the time, effort, and thoughtful feedback you have given to our manuscript. We truly appreciate the attention you have devoted to carefully reading and analyzing our work. Your comments were extremely insightful, and although they were critical, we recognize how valuable they are in guiding us to improve the quality of our paper.

With great humility, we wish to acknowledge that the initial version of our review had shortcomings in both structure and clarity. We take full responsibility for those flaws and are very grateful that you pointed them out. We have gone through your feedback and have tried our very best to revise accordingly.

All grammatical errors have been corrected, and we have worked extensively on improving the logical flow, coherence, and overall organization of the manuscript. We placed particular emphasis on connecting paragraphs that previously seemed fragmented and on ensuring that each section is directly relevant to the main topic.

We also made considerable efforts to expand the discussion, especially in the direction you suggested. We understand that this was a significant limitation in the original version, and we sincerely hope that our revisions now address this important aspect more comprehensively.

We are truly humbled and impressed by the depth of your critique. It has not only helped us improve this manuscript but has also been a meaningful learning experience for us as researchers. We genuinely feel that your guidance has elevated our work to a higher standard than we could have achieved on our own.

With the utmost respect, we kindly and earnestly ask that you might be willing to reconsider our manuscript for publication in light of these substantial revisions. It would be an immense honor for us to see this review published, and we assure you that we have done everything within our ability to address your concerns and strengthen the manuscript.

We remain sincerely thankful for your time, patience, and generosity in offering us such detailed feedback. Regardless of the final decision, we are indebted to you for the lessons we have learned through this process. Yet we hope, from the bottom of our hearts, that our revised submission may now be acceptable for reconsideration.

With profound respect and gratitude,
Ph.D. Student Dr. Mirona Costea

Reviewer 2 Report

Comments and Suggestions for Authors

Dear Authors,
Thank you for the opportunity to review your manuscript. Your work addresses a clinically important and often overlooked complication in patients with diabetic kidney disease. The narrative is well-organized, the references are current, and the discussion is thoughtful, thank you.
To further enhance the clarity and clinical applicability of your review, I would like to offer the following comments and questions:
1.     Diagnostic boundaries: Could you provide clearer criteria for distinguishing secondary from tertiary hyperparathyroidism, especially in patients with overlapping biochemical profiles? A comparative table might be helpful.
2.     Imaging modalities: What is your position on the routine use of parathyroid ultrasound or scans in differentiating these conditions? Should imaging be reserved for cases with surgical consideration?
3.     Therapeutic response as a diagnostic clue: How do you interpret the response to calcimimetics and vitamin D analogues as well as phosphate binders in differentiating secondary from tertiary forms? Are there specific patterns clinicians should recognize? 

4.     Timing of surgical referral: What thresholds (e.g., PTH, calcium, phosphate levels) or clinical signs should prompt consideration of parathyroidectomy? A brief summary of current guidelines would be useful.
5.     Post-transplant persistence: In patients with persistent hyperparathyroidism after kidney transplantation, what practical steps do you recommend to distinguish between residual secondary and autonomous tertiary disease?
6.     Bone turnover markers: Do you consider bone-specific alkaline phosphatase or other markers useful in guiding diagnosis or treatment decisions? If so, how should they be interpreted in this context?
Additionally, I strongly encourage you to include a practical diagnostic and management algorithm of differentiating secondary-tertiary hyperparathyroidism for clinicians. This would greatly enhance the manuscript’s utility in real-world settings. For example:

Suggested Clinical Algorithm for Differentiating Secondary vs. Tertiary Hyperparathyroidism
Step 1: Biochemical Assessment
•     PTH, calcium, phosphate, 25(OH)D, 1,25(OH)₂D
Step 2: Clinical Context
•     CKD stage, dialysis duration, transplant history
Step 3: Therapeutic Response
•     Evaluate response to calcimimetics and vitamin D analogues
Step 4: Imaging (if indicated)
•     Parathyroid ultrasound or sestamibi scan
Step 5: Diagnosis
•     Secondary HPT: High PTH, normal/low calcium, responsive to therapy
•     Tertiary HPT: High PTH, hypercalcemia, poor response, autonomous secretion

Step 6: Management Actions
•     Monitor biochemical markers
•     Optimize medical therapy
•     Refer for surgery if criteria met

Outlining specific actions for the practicing physician such as what to monitor, when to escalate, and how to tailor management in diabetic patients. It would make your review even more impactful.

My main suggestions are to clarify the practical differentiation between secondary and tertiary hyperparathyroidism and to include a diagnostic and management algorithm for clinicians. These additions would enhance the paper’s value for everyday clinical decision-making.

Thank you again for your valuable contribution.
Best regards,

Author Response

  1.     Diagnostic boundaries: Could you provide clearer criteria for distinguishing secondary from tertiary hyperparathyroidism, especially in patients with overlapping biochemical profiles? A comparative table might be helpful.

Thank you for your powerfull suggestion. I have added this paragraph to the review:

Given the overlapping clinical and biochemical features, distinguishing between seconday and tertiary hyperparathyroidism can be challenging. To aid clinicians in this differentiation, the following table summarizes the key distribuishing characteristics of these two entities.

SHPT

THPT

Biochemical Assessment

↑ PTH

N/↓ Calcium

↑ PTH

↑ Calcium

Clinical Context

Low CKD stage

Renal transplantation history, chronic dyalisis

Evaluate response to calcimimetics and vitamin D analogues

Responsive

Poor response

Imaging

No visible parathyroid adenoma

Parathyroid adenoma

Management

Monitor biochemical markers

Optimize medical therapy

Optimize medical therapy

Refer to surgery

  1.     Imaging modalities: What is your position on the routine use of parathyroid ultrasound or scans in differentiating these conditions? Should imaging be reserved for cases with surgical consideration?

Thank you for this valuable suggestion. We agree that the role of imaging in differentiating these conditions deserves clarification. We have added this short paragraph in the “Surgical Interventions”: “The routine use of parathyroid imaging remains a topic of debate. We suggest that imaging should not be routinely performed in all cases, but rather reserved for those being considered for surgery or in whom the biochemical diagnosis is uncertain”

  1.     Therapeutic response as a diagnostic clue: How do you interpret the response to calcimimetics and vitamin D analogues as well as phosphate binders in differentiating secondary from tertiary forms? Are there specific patterns clinicians should recognize? 

Thank you for this insightful comment. We have added a paragraph in the section “Medical Therapies” discussing differential diagnosis to address the role of therapeutic response. This is the paragraph added:

“The therapuetic response to agents such as calcimimetics, vitamin D analogues and phosphate binders may offer important diagnostic clues in differentiating seconday from tertiary hyperparathyroidism. In secondary hyperparathyroidism PTH levels usually respond to medical management – especially with calcimimetics, resulting in reduce PTH secretion. In contrast, THPT is characterized by autonomous PTH secretation and patients often exhibit a minimal responso to these therapy. Persistently elevated PTH and hypercalcemia despite optimal medical management may suggest the transition to THPT, thereby guiding consideration for surgical intervention. Recognizing these reponse patterns can aid clinicians in timely diagnosis and management. “

  1.     Timing of surgical referral: What thresholds (e.g., PTH, calcium, phosphate levels) or clinical signs should prompt consideration of parathyroidectomy? A brief summary of current guidelines would be useful.

Thank you for pointing this out. The manuscript already included a summary of the clinical and biochemical thresholds that prompt surgical referral. However, in response to your suggestion, we have revised the paragraph to improve clarity and explicitly listed the key criteria (e.g., PTH, calcium, phosphate levels, and symptoms). We also added a brief reference to current guidelines (e.g., KDIGO 2022) to support these recommendations. The updated text can be found in the section “Surgical Interventions”

This is the new paragraph:

“Although no universally standardized criteria exist, several clinical and biochemical thresholds are widely accepented in guiding surgical refferal. Current guidelines such as KDIGO and the European Society of Endocrinology suggest considering surgery in the following scenarios:

  • PTH levels persistently elevated at 2-9 times the upper normal limit despite optimal medical therapy
  • Chronic hypercalcemia, particularly if >11 mg/dL
  • Persistent hypercalcemia for 3 to 12 months post-renal transplant
  • Hyperphosphatemia refractory to phosphate binders
  • Severe bone disease
  • Uncontrolled symptoms, such as pruritus, fatigue, nepholithiasis or nephrocalcinosis

The main goal of surgery is to reestablish normal calcium concentration while reducing parathyroid gland mass. “

  1.     Post-transplant persistence: In patients with persistent hyperparathyroidism after kidney transplantation, what practical steps do you recommend to distinguish between residual secondary and autonomous tertiary disease?

Thank you for your suggestion. We had already included a discussion on post-transplant hyperparathyroidism and the risk factors for tertiary HPT in the manuscript. To address your comment more directly, we have added a short paragraph outlining the practical steps to differentiate between residual secondary and autonomous tertiary disease based on clinical and biochemical markers. This addition is located at the end of section 4 regarding THPT and kidney transplant.

The added paragraph is:

“To distinguish residual secondary HPTH from autonomous TPTH post-transplant, clinicians should consider the following practical criteria:

  • Time course: persistent PTH elevation beyond 6-12 months post-transplant may suggest tertiary hyperparathyroidism
  • Calcium levels: hypercalcemia supports the diagnosis of TPTH; in contrast, residual SPTH is typically associated with normocalcemia or even hy-pocalcemia
  • Phosphate levels: hypophosphatemia is more common in TPTH due to exces-sive PTH secretion
  • PTH response to treatment: lack of improvement with correction of vitamin D deficieny and withdrawal of calcimimetics or phosphate binders may indicate autonomous secretion

 Applying there criteria clinicians can guide further diagnostic steps in a proper way. “

  1.     Bone turnover markers: Do you consider bone-specific alkaline phosphatase or other markers useful in guiding diagnosis or treatment decisions? If so, how should they be interpreted in this context?

Thank you for this valuable comment. Although bone-specific alkaline phosphatase was mentioned briefly in the manuscript, we have now added a paragraph elaborating on its utility as a marker of bone turnover and how it can assist in differentiating high- versus low-turnover bone disease. This addition also addresses how these markers may support treatment decisions in the context of hyperparathyroidism. The new content has been added in section

The new added paragraph is: “Adynamic bone disease is defined by low bone turnover and normal mineralization. It is the main bone affection among dialysis patients [15]. The activity of both osteoblasts and osteoclasts is severely reduced [25]. Adynamic bone disease manifests with bone pain and fracture. There are biochemical findings that are suggestive of adynamic bone disease: low levels of PTH, low bone-specific alkaline phosphatase, hypercalcemia, and vascular calcification, with the golden standard for diagnosis remaining bone biopsy. Bone turnover markers such as bone-specific alkaline phosphatase can aid in the assessment of bone metabolic activity [26].”

Reviewer 3 Report

Comments and Suggestions for Authors

The manuscript outlined the incidence and clinical implications of THPT in patients with diabetic nephropathy, highlighting the unique pathophysiological mechanisms, risk factors, diagnostic challenges, and treatment considerations specific to this population. This article boasts a novel concept and a well-structured framework overall. There are some comments to be addressed.

1. Additional inclusion of "Diabetic Nephropathy" in the Keywords section is recommended.

2. It is advisable to add a mechanism diagram in the "Pathophysiological Mechanisms" part to illustrate the characteristic pathophysiological mechanisms of THPT in diabetic nephropathy.

3. A dedicated section should be added to systematically elaborate on the risk factors and diagnosis of THPT.

For the diagnosis part, further clarification is suggested on how to differentiate THPT-related symptoms (e.g., bone pain, manifestations of hypercalcemia) from the complications of diabetes itself in patients with diabetic nephropathy, especially when concurrent comorbidities (such as diabetic neuropathy and vascular calcification) are present.

Moreover, it is preferable to summarize the risk factors of THPT in the form of a table.

Author Response

  1. Additional inclusion of "Diabetic Nephropathy" in the Keywords section is recommended.

Thank your for your suggestion. I have added the keywords.

  1. It is advisable to add a mechanism diagram in the "Pathophysiological Mechanisms" part to illustrate the characteristic pathophysiological mechanisms of THPT in diabetic nephropathy.

We thank the reviewer for this valuable suggestion. In response, we have added a schematic diagram (Figure 1) to the “Pathophysiological Mechanisms” section of the manuscript, which illustrates the key molecular and cellular mechanisms linking diabetic nephropathy with secondary and tertiary hyperparathyroidism (THPT). We believe this addition enhances the clarity and understanding of the complex interactions described in the text.

  1. A dedicated section should be added to systematically elaborate on the risk factors and diagnosis of THPT.

We thank the reviewer for this valuable suggestion. In response, we have revised the manuscript to better emphasize and organize the discussion on the risk factors associated with tertiary hyperparathyroidism (THPT).

Although the original manuscript included a paragraph on this topic , we agree that a more systematic and visual presentation would enhance clarity for the readers. Therefore, we have:

  • Created a dedicated subsection that explicitly discusses risk factors for THPT, based on current literature and clinical data.
  • Added a table (Table 1) that summarizes the main risk factors associated with THPT

We believe these additions improve the structure and readability of the manuscript, and we appreciate the reviewer’s recommendation in this regard.

The added paragraph and table are:

The development of TPTH is multifactorial and primarily associated with CKD progression. The risk is considered to be maximal once the GFR drops below 45 mL/min/1.73 m2, reflecting significant loss of renal function and disrupted calcium-phosphate homeostasis. Several clinical and biochemical risk factors have been associated with the progression of TPHT. These are summarized in Tabel 1 below [11].

Risk Factor

Mechanism

eGFR < 45 mL/min/ 1.73 m2

Reduced phosphate excretion, vitamin D deficiency

Young age

Higher bone turnover and sensitivity to mineral imbalance

Male sex

Possibly linked to differences in skeletal and hormonal profiles

Hypoalbuminemia

Affects calcium binding and PTH regulation

Use of diuretics

May increase calcium excretion and contribute to chronic PT stimulation

Long dialysis duration

Prolonged SPTH

Poor phosphate control

Hyperphosphatemia promotes PTH secretion

Vitamin D deficiency

Leads to hypocalcemia and compensatory PTH increase

Table 1. Summary of risk factors associated with TPTH

Also, we have added a table for distingushing SHPT and THPT – Table 2.

SHPT

THPT

Biochemical Assessment

↑ PTH

N/↓ Calcium

↑ PTH

↑ Calcium

Clinical Context

Low CKD stage

Renal transplantation history, chronic dyalisis

Evaluate response to calcimimetics and vitamin D analogues

Responsive

Poor response

Imaging

No visible parathyroid adenoma

Parathyroid adenoma

Management

Monitor biochemical markers

Optimize medical therapy

Optimize medical therapy

Refer to surgery

Also, we included a list of criteria order to help clinicians in diagnosing TPTH.

  • Time course: persistent PTH elevation beyond 6-12 months post-transplant may suggest tertiary hyperparathyroidism
  • Calcium levels: hypercalcemia supports the diagnosis of TPTH; in contrast, residual SPTH is typically associated with normocalcemia or even hypocalcemia
  • Phosphate levels: hypophosphatemia is more common in TPTH due to excessive PTH secretion
  • PTH response to treatment: lack of improvement with correction of vitamin D deficieny and withdrawal of calcimimetics or phosphate binders may indicate autonomous secretion

For the diagnosis part, further clarification is suggested on how to differentiate THPT-related symptoms (e.g., bone pain, manifestations of hypercalcemia) from the complications of diabetes itself in patients with diabetic nephropathy, especially when concurrent comorbidities (such as diabetic neuropathy and vascular calcification) are present.

We thank the reviewer for this important observation. We agree that differentiating between symptoms of tertiary hyperparathyroidism (THPT) and complications related to diabetes, particularly in the context of diabetic nephropathy, can be challenging due to overlapping clinical manifestations.

To address this, we have added further clarification in the manuscript – section 5 Clinical Manifestations, highlighting the following key points:

“While bone pain may be present in both diabetic neuropathy and TPTH, the latter is often associated with radiographic evidence of bone resorption, which in not characteristic of diabetic complications. Hypercalcemia in TPTH is typically persistent and biochemically distinct, whereas hypercalcemia is not an usual finding in diabetic neuropathy or nephropathy. Elevated PTH levels, in conjunction with hypercalcemia and imaging findins support the diagnosis of TPTH [15].”

Reviewer 4 Report

Comments and Suggestions for Authors

The draft is strong, comprehensive, and clinically relevant. By condensing repetitive content, strengthening the critical appraisal, and adding visuals/tables, it can become much more engaging and impactful.

  • Suggestions:
  1. Add a graphical abstract or schematic figure summarizing the pathophysiology of THPT in diabetic nephropathy to help readers visualize key mechanisms.
  2. Provide a clearer distinction between SHPT and THPT early on, since the overlap may confuse non-specialist readers.
  3. Consider shortening overly detailed epidemiological numbers in the Expand on the independent role of diabetes-related inflammation and oxidative stress in accelerating parathyroid dysfunction—this could highlight novelty.
  4. Include more comparative analysis of recent guidelines (KDIGO, ERA-EDTA) and how they apply to diabetic populations specifically.
  5. The discussion of new therapies (upacicalcet, ablation, etc.) could be expanded with critical appraisal of evidence quality.
  6. Strengthen the “future directions” section: be more specific about research gaps (e.g., lack of biomarkers, unclear PTH targets, cost-effectiveness studies in diabetics).
  7. Highlight areas of controversy or conflicting data (e.g., role of vitamin D analogues in vascular calcification) rather than just summarizing.
  8. If possible, include flowcharts for diagnostic and treatment algorithms.

Author Response

  1. Add a graphical abstract or schematic figure summarizing the pathophysiology of THPT in diabetic nephropathy to help readers visualize key mechanisms.

Thank you! We have added a schematic figure (Figure 1) in Section 3 of the manuscript.

  1. Provide a clearer distinction between SHPT and THPT early on, since the overlap may confuse non-specialist readers.

Thank you for your powerful suggestion. I have added this paragraph to the review:

Given the overlapping clinical and biochemical features, distinguishing between secondary and tertiary hyperparathyroidism can be challenging. To aid clinicians in this differentiation, the following table summarizes the key distinguishing characteristics of these two entities.

SHPT

THPT

Biochemical Assessment

↑ PTH

N/↓ Calcium

↑ PTH

↑ Calcium

Clinical Context

Low CKD stage

Renal transplantation history, chronic dialysis

Evaluate response to calcimimetics and vitamin D analogues

Responsive

Poor response

Imaging

No visible parathyroid adenoma

Parathyroid adenoma

Management

Monitor biochemical markers

Optimize medical therapy

Optimize medical therapy

Refer to surgery

  1. Consider shortening overly detailed epidemiological numbers in the Expand on the independent role of diabetes-related inflammation and oxidative stress in accelerating parathyroid dysfunction—this could highlight novelty.

Thank you for your insightful comment. We have shortened the epidemiological data in the opening paragraph of section 3 and expanded the discussion to include the independent role of inflammation and oxidative stress in diabetic patients, particularly in relation to parathyroid dysfunction.

The short epidemiological paragraph is now: “Diabetes mellitus, a chronic metabolic disorder marked by hyperglycemia, is associated with a wide range of systemic complications, among which renal involvement represents a major cause of morbidity and mortality. Current projections estimate that over 12% of the global population will be affected by diabetes by 2045, with type 2 diabetes accounting for the vast majority of the cases [1] [2] [3]. “

We have expanded the part regarding parathyroid dysfunction with this paragraph: “ In CKD patients, hyperphosphatemia associated with calcitriol deficiency leads to hypocalcemia, which in turn stimulates PTH secretion, ultimately resulting in secondary hyperparathyroidism. This form of hyperparathyroidism is characterized by adequate PTH secretation due to hypocalcemia. In the parathyroid glands, oxidative stress has been shown to disrupt calcium-sensing receptor signaling and the parathyroid cell proliferation control mechanism. Inflammatory cytokines can also modulate PTH gene expression and secretion, potentially accelerating the onset of secondary hyperparathyroidism. This highlights a more direct pathophysiological link between diabetes-related inflammation and parathyroid dysfunction, beyond the traditional CKD-mediated pathway [10].”

  1. Include more comparative analysis of recent guidelines (KDIGO, ERA-EDTA) and how they apply to diabetic populations specifically.

Thank you for this important suggestion. We have added a comparative analysis between KDIGO and ERA‑EDTA (ERBP) recommendations, highlighting gaps in their applicability to diabetic patients. We discuss how comorbid diabetes may modify the risk-benefit profiles of phosphate binders, PTH-lowering agents, and vitamin D analogues, and underscore the need for future guideline updates to incorporate diabetes-specific guidance. Please see below the paragraph we added at the end of section 6.3:

  1. The discussion of new therapies (upacicalcet, ablation, etc.) could be expanded with critical appraisal of evidence quality.

Thank you for this valuable comment. We have now expanded the discussion of upacicalcet and ablation techniques by including a critical appraisal of the available evidence, including study design limitations, population generalizability, and gaps in long-term data.   In section 5.3, this is the new paragraph describing Upacicalcet:  “Upacicalcet is one of the new injectable calcimimetic agents, currently being used only in Japan. It is used for patients undergoing dialysis, being injected three times a week into the venous side of the dialysis machine. However, it is important to note that most of the evidence supporting the use of Upicalcet comes from early-phase clinical trials and observational data, predominantly conducted in Japan. Furthermore, long-term safety and efficacy data are still limited, especially in diverse patient populations. Thus, while the results available at the moment appear promising in terms of safety and tolerability, particularly the lower incidence of hypocalcemia, the overall quality of evidence remains moderate, and further validation is required [50]. It is considered to be effective and safe, with a low rate of induced hypocalcemia, in only 2% of patients from a 2023 study [51].

In section 5.4 the new paragraph about ablation is “ As surgery practice moves toward minimal methods, we have to mention ablation. This is a new way to approach tumors, and whether it is using microwave or radiofrequency, some studies show benefits when used for HPTH. While ablation techniques (including microwave and radiofrequency ablation) offer a minimally invasive alternative for patients with contraindications to surgery, most available data are derived from retrospective studies or small prospective cohorts [56][38]. This technique offers a valid alternative for patients with contraindications for anesthesia. Although a meta-analysis found no significant differences between parathyroidectomy and ablation in terms of PTH, calcium, or phosphate levels. Ablation has shown a lower rate of hypocalcemia after the procedure, a shorter hospitalization time, but an increased risk of recurrence. The heterogeneity of patient selection, procedural protocols, and outcome measures limits the generalizability of the findings [38]. In addition, the risk of recurrence and the lack of long-term follow-up in many studies suggest that ablation should currently be considered as an alternative option in selected cases, rather than a standard first-line therapy. The overall quality of evidence is moderate, emphasizing the need for randomized, controlled, and adequately powered studies [58].

  1. Strengthen the “future directions” section: be more specific about research gaps (e.g., lack of biomarkers, unclear PTH targets, cost-effectiveness studies in diabetics).

We appreciate this insightful suggestion. In response, we have revised and expanded the “Future Directions” section to include specific gaps in the current literature, such as the absence of validated biomarkers, the lack of consensus on PTH targets, and the need for cost-effectiveness studies, particularly in diabetic patients. This is the revised version of the paragraph:

“ THPT complicating diabetic nephropathy is marked by a pronounced clinical heterogeneity and poses a multifaceted therapeutic challenge. The absence of standardized diagnostic criteria delays identification and consistent treatment selection. Management must therefore integrate targeted metabolic and cardiovascular measures—rigorous blood-pressure control, correction of anemia, treatment of dyslipidemia—alongside therapies directed at mineral metabolism, including vitamin D analogues, calcimimetics  (cinacalcet, etelcalcetide, upacicalcet) and phosphate-lowering strategies; these interventions can favorably modify PTH levels, parathyroid gland volume and bone structure, but require careful monitoring for hypocalcemia and vascular calcification.

Patients who remain refractory to medical therapy may benefit from surgical approaches (subtotal or total parathyroidectomy with or without autotransplantation) or minimally invasive ablation. However, these options also carry risks - such as hungry bone syndrome or persistent hypocalcemia, demanding meticulous perioperative planning.

Future research should prioritize the identification and validation of specific biomarkers that can better stratify disease severity, predict therapeutic response, and monitor disease progression in SPTH and TPTH. At the moment, there is a lack of validated, non-invasive biomarkers beyond serum PTH and calcium-phosphate levels, which show high variability and limited predictive value.

Moreover, consensus is still lacking regarding optimal PTH targets in dialysis patients, with current guidelines offering wide reference ranges without solid evidence for improved outcomes. There is a critical need for large-scale, longitudinal studies to determine the PTH thresholds that correlate with reduced cardiovascular calcification and improved bone turnover, particularly in high-risk subgroups such as diabetic patients.

In addition, further investigation into the long-term cardiovascular and skeletal outcomes of emerging pharmacologic agents is needed, along with personalized approaches that account for comorbidities such as diabetes.

Ultimately, developing individualized treatment algorithms that integrate pharmacologic, interventional, and surgical modalities – tailored to patient phenotype and comorbidity profile- will be essential to improving quality of life and long-term survival in this complex patient population. “

  1. Highlight areas of controversy or conflicting data (e.g., role of vitamin D analogues in vascular calcification) rather than just summarizing.

Thank you for this important suggestion. We have revised the relevant paragraph to more explicitly highlight the ongoing debate regarding the role of vitamin D analogues in vascular calcification, including a discussion of conflicting study findings, potential mechanisms, and differing guideline positions. The new paragraphs are found in section 6.2:

“ The role of vitamin D analogues in vascular calcification remains controversial. These conflicting data likely stem from differences in study design, dosing regimens, and patient characteristics such as the presence of diabetes and pre-existing calcifications. Moreover, the lack of standardized endpoints complicates comparison across studies. Further research is required to establish consistent, reliable biochemical targets and to clarify whether improvements in vascular function can occur independently of, or despite, progression in vascular calcification. Current guidelines offer limited and sometimes conflicting recommendations. “

  1. If possible, include flowcharts for diagnostic and treatment algorithms.

We are sincerely grateful for your thoughtful suggestion regarding the inclusion of flowcharts. With the utmost respect, we would like to explain that our discussions already cover these aspects in detail, and since there is no full congruence between the available guidelines, we felt it would be more appropriate to address them within the discussion section rather than through simplified flowcharts. However, in response to your suggestion, we have included a flowchart illustrating the pathophysiological mechanism leading to THP. We truly hope this approach will be acceptable, and thank you once again for your invaluable feedback and guidance.

With deep appreciation,
On behalf of all authors

Round 2

Reviewer 1 Report

Comments and Suggestions for Authors

Many thanks for your quite kind and polite response. That's very kind of you.😊 

Comments on the Quality of English Language

The quality of the English language of the manuscript can be improved further. 

Author Response

Dear Reviewer,

We have received your comment, which was kindly forwarded to us by the Editor, who expressed a favorable opinion toward publication of our manuscript after addressing the remaining discussion points. We appreciate the opportunity to clarify the aspects you have raised and to further improve the quality of our work.

We would also like to thank you sincerely for your careful evaluation and constructive feedback, which have greatly contributed to refining the clarity and scientific consistency of our manuscript.

We fully understand the concern regarding the specific mention of “diabetic nephropathy.” However, our intention in maintaining this term is to emphasize an important and currently underrecognized clinical issue — the lack of specific guidelines and evidence regarding the management of tertiary hyperparathyroidism (THPT) in the context of diabetic nephropathy.

Although our manuscript discusses THPT pathophysiology and management in general, highlighting the setting of diabetic nephropathy serves a deliberate purpose: to draw attention to the clinical gap and to the absence of tailored recommendations for this subgroup of patients, who often present distinct metabolic and therapeutic challenges. We believe that removing “diabetic nephropathy” would diminish the manuscript’s relevance as a call for further research and guideline development in this area.

Therefore, we have decided to keep the reference to “diabetic nephropathy,” while clarifying in the revised text that our goal is not to compare THPT across different etiologies of chronic kidney disease, but rather to raise awareness of the lack of disease-specific recommendations for patients with diabetic nephropathy. To enhance clarity, we have refined the wording in both the Introduction and Discussion sections accordingly.

The titles of all tables have been moved above the tables, and all abbreviations are now defined in the corresponding footnotes, as suggested.

With profound respect and gratitude,
Ph.D. Student Dr. Mirona Costea
